# Dynasore Blocks Ferroptosis through Combined Modulation of Iron Uptake and Inhibition of Mitochondrial Respiration

**DOI:** 10.3390/cells9102259

**Published:** 2020-10-09

**Authors:** Laura Prieto Clemente, Malena Rabenau, Stephan Tang, Josefina Stanka, Eileen Cors, Jenny Stroh, Carsten Culmsee, Silvia von Karstedt

**Affiliations:** 1Department of Translational Genomics, Center of Integrated Oncology Cologne-Bonn, University Hospital of Cologne, 50931 Cologne, Germany; lprietoc@uni-koeln.de (L.P.C.); s0jostan@uni-bonn.de (J.S.); j.stroh@uni-koeln.de (J.S.); 2CECAD Cluster of Excellence, University of Cologne, 50931 Cologne, Germany; 3Institute for Pharmacology and Clinical Pharmacy, Biochemical-Pharmacological Center Marburg, University of Marburg, 35032 Marburg, Germany; malena.rabenau@pharmazie.uni-marburg.de (M.R.); tangste@staff.uni-marburg.de (S.T.); Cors@students.uni-marburg.de (E.C.); culmsee@uni-marburg.de (C.C.); 4Center for Mind, Brain and Behavior- CMBB, University of Marburg, 35032 Marburg, Germany; 5Center for Molecular Medicine Cologne, Medical Faculty, University Hospital of Cologne, 50931 Cologne, Germany

**Keywords:** ferroptosis, radical scavenger, dynasore

## Abstract

Ferroptosis is a form of regulated necrosis characterized by a chain-reaction of detrimental membrane lipid peroxidation following collapse of glutathione peroxidase 4 (Gpx4) activity. This lipid peroxidation is catalyzed by labile ferric iron. Therefore, iron import mediated via transferrin receptors and both, enzymatic and non-enzymatic iron-dependent radical formation are crucial prerequisites for the execution of ferroptosis. Intriguingly, the dynamin inhibitor dynasore, which has been shown to block transferrin receptor endocytosis, can protect from ischemia/reperfusion injury as well as neuronal cell death following spinal cord injury. Yet, it is unknown how dynasore exerts these cell death-protective effects. Using small interfering RNA suppression, lipid reactive oxygen species (ROS), iron tracers and bona fide inducers of ferroptosis, we find that dynasore treatment in lung adenocarcinoma and neuronal cell lines strongly protects these from ferroptosis. Surprisingly, while the dynasore targets dynamin 1 and 2 promote extracellular iron uptake, their silencing was not sufficient to block ferroptosis suggesting that this route of extracellular iron uptake is dispensable for acute induction of ferroptosis and dynasore must have an additional off-target activity mediating full ferroptosis protection. Instead, in intact cells, dynasore inhibited mitochondrial respiration and thereby mitochondrial ROS production which can feed into detrimental lipid peroxidation and ferroptotic cell death in the presence of labile iron. In addition, in cell free systems, dynasore showed radical scavenger properties and acted as a broadly active antioxidant which is superior to N-acetylcysteine (NAC) in blocking ferroptosis. Thus, dynasore can function as a highly active inhibitor of ROS-driven types of cell death via combined modulation of the iron pool and inhibition of general ROS by simultaneously blocking two routes required for ROS and lipid-ROS driven cell death, respectively. These data have important implications for the interpretation of studies observing tissue-protective effects of this dynamin inhibitor as well as raise awareness that off-target ROS scavenging activities of small molecules used to interrogate the ferroptosis pathway should be taken into consideration.

## 1. Introduction

Recently, ferroptosis was described as an iron-dependent form of regulated necrosis [1]. One characteristic hallmark of ferroptosis is the accumulation of peroxidized lipids, which are thought to destabilize the plasma membrane bilayer [2]. Once lipids have been peroxidized, this can trigger a radical chain reaction within lipid bilayers [3]. Lipophilic antioxidants, including vitamin E derivatives [4], ferrostatin-1 and liproxstatin-1 [5], but also endogenous ubiquinol generated from ubiquinone through ferroptosis suppressor protein 1 (Fsp1) activity [6,7], can prevent ferroptosis by acting as radical-trapping agents (RTAs) within membranes [8]. Importantly, to prevent constitutive membrane peroxidation, glutathione peroxidase 4 (GPX4) reduces lipid peroxides and is thereby central to protecting cells from ferroptosis [1,5]. GPX4 in turn requires glutathione (GSH) as an electron donor, of which the synthesis depends on intracellular cysteine availability. Among other strategies, this is ensured by the cystine/glutamate antiporter subunit xCT, which thereby protects cells from ferroptosis [1].

Oxidation of membrane lipids can occur spontaneously through a process requiring divalent iron to promote a Fenton reaction, generating hydroxyl radicals which then react with membrane lipids. Alternatively, lipid peroxidation can be catalyzed by the iron-containing lipoxygenases (LOXs), which target polyunsaturated fatty acids (PUFAs) for peroxidation [9]. In both cases, ferroptosis requires intracellular divalent iron pools for lipid peroxidation, and, consequently, chelation of iron by deferoxamine (DFO) rescues cells from ferroptosis [1]. Iron can be transported into the cell complexed with transferrin via binding to transferrin receptor (TFR1), followed by endocytosis [10]. Supporting the importance of this import route of ferroptosis-active iron, both transferrin and TFR1 were shown to promote ferroptosis [11,12]. Interestingly, transferrin endocytosis is dependent on the GTPase dynamin 1 under steady-state turnover [13].

The dynamin family of large GTPases fulfils important functions in cellular membrane shape regulation, including endocytosis of surface receptors via clathrin-coated pits and their membrane scission. Therefore, small molecules targeting dynamins have been frequently used for the investigation of these processes in many studies worldwide. For example, dynasore, an inhibitor developed against dynamin 1 and 2 [14], was described to successfully block transferrin endocytosis and uptake. Intriguingly, dynasore has been shown to protect neurons from cell death after spinal cord injury [15] as well as ischemia/reperfusion injury in mouse hearts [16], a type of tissue injury that was also blocked by the ferroptosis inhibitor ferrostatin-1 [17,18]. However, an explanation of how dynamin inhibitors might protect from tissue damage and cell death has remained elusive. Therefore, we set out to determine the mechanism by which dynasore might protect from cell death.

## 2. Materials and Methods

### 2.1. Reagents

Erastin (Biomol, Hamburg, Germany), dynasore (Sigma Aldrich, Taufkirchen, Germany), RSL3 (Selleckchem, Munich, Germany), ferrostatin-1 (Sigma Aldrich, Taufkirchen, Germany), Nec1s (Abcam, Berlin, Germany), zVAD (ENZO, Lörrach, Germany), H_2_O_2_ 30% (Sigma Aldrich, Taufkirchen, Germany), 2,2-diphenyl-1-picrylhydrazyl (DPPH; Cayman Chemical, MI, USA), ADP (Sigma Aldrich, Taufkirchen, Germany) oligomycin (Cayman Chemical, MI, USA), FCCP (Sigma Aldrich, Taufkirchen, Germany), antimycin A (Sigma Aldrich, Taufkirchen, Germany), rotenone (Sigma Aldrich, Taufkirchen, Germany), MTT (Sigma Aldrich, Taufkirchen, Germany), L-glutamine (Merck KGaA, Darmstadt, Germany) and 2-DG (Carl Roth GmbH, Karlsruhe, Germany).

### 2.2. Cell Lines and Culture Conditions

Human non-small cell lung cancer (NSCLC) cell lines (H441, A549, H460, H727, H520, H332m) were kindly provided by Prof. Julian Downward and the small cell lung cancer cell line (SCLC) H82 was kindly provided by Prof. Roman Thomas. They were cultured in a humidified 37 °C atmosphere containing 10% CO_2_ in RPMI 1640 medium (Thermo Fisher, Darmstadt, Germany) supplemented with 10% fetal bovine serum (FBS) (Sigma Aldrich, Taufkirchen, Germany) and 1000 U/mL of both penicillin and streptomycin (Sigma Aldrich, Taufkirchen, Germany), and were tested for mycoplasma at regular intervals (mycoplasma barcodes, Eurofins Genomics, Ebersberg, Germany). HT22 cells (kindly provided by David Schubert, Cellular Neurobiology Laboratory, Salk Institute for Biological Studies, La Jolla, CA, USA) were grown in Dulbecco’s modified Eagle’s medium (DMEM, Capricorn Scientific GmbH, Ebsdorfergrund, Germany) supplemented with 10% heat-inactivated FBS (Merck KGaA, Darmstadt, Germany), 100 U/mL penicillin, 100 mg/mL streptomycin (Capricorn Scientific GmbH, Ebsdorfergrund, Germany) and 2 mM L-glutamine (Merck KGaA, Darmstadt, Germany).

### 2.3. Transfection with Small Interfering RNA (siRNA)

For dynamin 1 and 2 knockdowns, 200 µL Opti-MEM (Gibco, Darmstadt, Germany) and 1.5 µL Dharmafect Reagent I (Dharmacon, Cambridge, UK) were mixed and used per well of a 6-well plate and were incubated for 5–10 min at room temperature. Next, 2.2 µL per 200 µL of siRNA (stock 20 mM) (Dharmacon, Cambridge, UK) were added to the mixture and incubated for 30 min at room temperature. Then, 200 µL of the mixture were added to each well of a 6-well plate and 300,000 cells were plated on top in 1 mL media. Knockdowns were incubated for 48–72 h, as indicated.

### 2.4. Cell Viability Assays

To determine viability 10,000 cells were plated in 100 µL media in each well of a 96-well plate. Compounds were added 1 day post-plating and incubated for 24 h. Then, 5 µl Cell Titer Blu reagent (Promega, Walldorf, Germany) was added to each well and cells were incubated for 1.5 h at 37 °C in atmosphere containing 10% CO_2_. The fluorescent substrate was measured using a plate reader (EnSight™ Multimode Microplate Reader, PerkinElmer, Rodgau, Germany). Metabolic activity as an indicator of cell viability was quantified using the MTT assay, as done previously [19]. Viable and metabolically active cells converted 3-(4,5-dimethylthiazol-2-yl)-2,5-diphenyltetrazolium bromide (MTT, Merck KGaA, Darmstadt, Germany), which was added at a concentration of 2.5 mg/mL for 1 h at 37 °C to the culture medium, into purple formazan. Absorbance was measured at 570 nm versus 630 nm with FluoStar (BMG Labtech, Ortenberg, Baden-Württemberg, Germany) after dissolving in DMSO (Carl Roth GmbH, Karlsruhe, Germany).

### 2.5. Fluorescence-Activated Cell Sorting (FACS) Assays

PI uptake/cell death assays: To determine cell death, 25,000 cells were plated in 500 µL media in each well of a 24-well plate. Compounds were added 1 day post-plating and cells were incubated for 48 h, followed by staining with propidium iodide (PI) (1 µg/mL) (Sigma Aldrich, Taufkirchen, Germany) in PBS (Thermo Fisher, Darmstadt, Germany) supplemented with 2% FBS. PI-positive cells were quantified by flow cytometry using an LSR-FACS Fortessa (BD Bioscience, Heidelberg, Germany) and FlowJo software (BD Bioscience, Heidelberg, Germany). CD71 staining: To determine CD71 surface levels, 25,000 cells were plated in each well of a 24-well plate. Cells were subjected to treatments as indicated. Cells were washed in PBS and stained for 30–40 min with the CD71 (Transferrin Receptor) Monoclonal Antibody (OKT9 (OKT-9)) (0.125 µg/test) FITC (Thermo Fisher, Darmstadt, Germany), and the Mouse IgG1 kappa Isotype Control (P3.6.2.8.1), FITC (Thermo Fisher, Darmstadt, Germany), (1 µg/test). Mean fluorescent intensity (MFI) was quantified by flow cytometry using an LSR-FACS Fortessa (BD Bioscience, Heidelberg, Germany) and FlowJo software (BD Bioscience, Heidelberg, Germany). Cell death of HT22 cells treated with dynasore and erastin was detected using the Annexin-V-FITC/PI Detection Kit (PromoCell, Heidelberg, Germany), followed by fluorescence-activated cell sorting (FACS, Guava easyCyte, Merck KGaA, Darmstadt, Germany). Annexin-V-FITC was excited at 488 nm and emission was detected through a 525 ± 30 nm bandpass filter. Propidium iodide was excited at 488 nm and fluorescence emission was detected using a 690 ± 50 nm bandpass filter. Flow cytometry data were collected from at least 5000 cells with at least three replicates per condition.

### 2.6. Western Blotting

After treatment, cells were washed in PBS, lysed in IP-lysis buffer (30 mM Tris-HCl (pH 7.4), 120 mM NaCl, 2 mM EDTA, 2 mM KCl, 1% Triton X-100, 1× COMPLETE protease-inhibitor cocktail) and frozen at −20 °C. After re-thawing, lysate concentrations were adjusted to equal protein concentrations using the bicinchoninic acid (BCA) protein assay (Biorad). Equal amounts of protein were mixed with a final concentration of 1× reducing sample buffer (Invitrogen) and 200 mM DTT (VWR). Samples were heated to 80 °C for 10 min, separated via gel electrophoresis and transferred to nitrocellulose membranes (Biorad, CA, USA) using the TurboBlotting system (Biorad, CA, USA). Membranes were blocked in PBS with 0.1% Tween 20 (PBST) (VWR, Langenfeld, Germany) with 5% (*w*/*v*) dried milk powder (AppliChem, Darmstadt, Germany) for at least 30 min. Next, membranes were incubated overnight at 4 °C with primary antibodies against CD71 (Santa Cruz Biotechnology, Dallas, TX, USA), β-actin (Sigma Aldrich, Taufkirchen, Germany), dynamin-1 (Cell signaling) and dynamin-2 (Sigma Aldrich, Taufkirchen, Germany), all diluted 1:1000 in PBST with 5% bovine serum albumin (BSA) (Thermo Fisher, Darmstadt, Germany ). After washing with PBST, membranes were incubated with horse radish peroxidase (HRP)-coupled secondary antibodies (Biotium, CA, USA) diluted 1:10,000 for at least 1 h at room temperature. After another washing step, bound antibodies were detected using chemiluminescent Classico Western HRP Substrate (Millipore, Taufkirchen, Germany) and X-ray films (Thermo Fisher, Taufkirchen, Germany).

### 2.7. Relative Intracelluar Iron Quantification

Relative levels of intracellular iron were determined using Phen Green SK diacetate (Thermo Fisher, Taufkirchen, Germany). First, 25,000 cells were plated in 500 µL in each well of a 24-well plate. During the last 30 min of treatment incubation, cells were washed at least three times with PBS 1x and Phen Green SK was resuspended in RPMI serum free and added to each well at 5 µM. Mean fluorescence intensity (MFI) was determined by flow cytometry using an LSR-FACS Fortessa (BD Bioscience, Heidelberg, Germany) and FlowJo software (BD Bioscience, Heidelberg, Germany). Flow cytometry data were collected from at least 5000 cells.

### 2.8. GSH Measurement

Relative levels of glutathione were determined using monochlorobimane (MCB) (Sigma Aldrich, Taufkirchen, Germany). First, 25,000 cells were plated in 500 µL in each well of a 24-well plate. During the last 30 min of treatment incubation, MCB was added to each well at 50 µM. Mean fluorescence intensity (MFI) was determined by flow cytometry using an LSR-FACS Fortessa (BD Bioscience, Heidelberg, Germany) and FlowJo software (BD Bioscience, Heidelberg, Germany). Flow cytometry data were collected from at least 5000 cells.

### 2.9. Lipid ROS Quantification

Lipid ROS levels were quantified by BODIPY-C11 (Invitrogen, Taufkirchen, Germany) staining. First, 25,000 cells were plated in 500 µL in each well of a 24-well plate. To stain cells, during the last 30 min of incubation BODIPY C11 was added at 5 µM to each well. Mean fluorescence intensity (MFI) was determined by flow cytometry using an LSR-FACS Fortessa (BD Bioscience, Heidelberg, Germany) and FlowJo software (BD Bioscience, Heidelberg, Germany). Flow cytometry data were collected from at least 5000 cells.

### 2.10. Mitochondrial ROS Quantification

Mitochondrial ROS levels were measured as described before for Lipid ROS. To stain cells, MitoSoX red (Thermo Fisher, Taufkirchen, Germany) was used at 2 µM per well. Flow cytometry data were collected from at least 5000 cells.

### 2.11. General Cellular ROS Quantification

Cellular ROS levels were measured as described before. To stain cells, H2DCFDF (Invitrogen, Taufkirchen, Germany) was used at 20 µM/well. Flow cytometry data were collected from at least 5000 cells.

### 2.12. Seahorse Assay in Intact Cells

To assess oxygen consumption rate (OCR) and extracellular acidification rate (ECAR) as measures of mitochondrial respiration and glycolysis, respectively, simultaneous real-time measurements were performed using the XF Extracellular Flux Analyzer (Agilent Technologies, Santa Clara, CA, USA), as previously described [19]. Briefly, HT22 cells were plated in XF96-well microplates (6000 cells per well, Seahorse Bioscience, Heidelberg, Germany) without seeding into the corner wells (background determination) and treated with the indicated compounds. After 16 h of treatment, the growth medium was replaced by ~180 μL of assay medium (containing 143 mM NaCl, 4.5 g/L glucose, 2 mM glutamine, 1 mM pyruvate, pH 7.35) and cells were incubated at 37 °C for 60 min. Three baseline measurements were recorded before adding the compounds. Oligomycin (ATP synthase inhibitor) (Merck KGaA, Darmstadt, Germany) was injected into port A (20 µL) at a final concentration of 3 µM, FCCP (uncoupling agent) (22.5 µL into port B) (Merck KGaA, Darmstadt, Germany) at a concentration of 0.5 µM, rotenone/antimycin A (complex I/III inhibitors) (25 µL into port C) (Merck KGaA, Darmstadt, Germany) at a concentration of 100 nM and 1 µM and 2-deoxyglucose (Carl Roth GmbH, Karlsruhe, Germany) (glycolysis inhibitor) at a concentration of 50 mM (27.5 µL into port D), respectively. Three measurements were performed after the addition of each compound (4 min mixing followed by 3 min detection). Data analysis was performed using XFe Wave software and visualized using GraphPad Prism software.

### 2.13. Seahorse Assay on Isolated Mitochondria

Mitochondria were isolated from the prefrontal cortex of Sprague Dawley rat brains, as described in Michels et al. [20]. Eight micrograms of purified mitochondria were used per well. Wells were treated with rising concentrations of dynasore for 1 h and then subjected to seahorse measurements, as described above. ADP (Sigma Aldrich) was injected into port A (20 µL) at a final concentration of 4 mM, oligomycin (Cayman Chemical, MI, USA) (22.5 µL into port B) at a final concentration of 2.5 µg/mL, FCCP (25 µL into port C) at a final concentration of 8 µM and antimycin A (27.5 µL into port D) at a final concentration of 4 µM.

### 2.14. DPPH Assay

To determine radical scavenging activity via the DPPH assay, dynasore or positive and negative control samples were prepared in 75% ethanol. Ninety microliters of 150 µM DPPH and 10 µL of the sample were incubated for 30 min in a 96-well plate in the dark. Absorbance was measured at 517 nm with a plate reader (SPARK 20M, TECAN, Baden-Württemberg, Germany). Radical scavenging activity was calculated using the following formula: (A_0_ − A_1_)/(A_0_) × 100.

### 2.15. Time-Lapse Cell Death Assays

Cells were plated in 96-well plates (5 × 10^3^) a day in advance. The next day, cells were stimulated as indicated. Dead cells were stained by adding 100 nM DRAQ7 (Thermofisher, Taufkirchen, Germany) to all wells. Cells were imaged for 48 h every 8 h and 3 images per well were captured using the Incucyte live-cell imaging system and automated quantification software (Essen BioScience, Royston, UK).

### 2.16. Quantification and Statistical Analysis

Statistical analysis was performed using GraphPad software (GraphPad Software Inc.). Two-tailed *t*-tests were performed for comparison between two conditions, two-way ANOVA and the Bonferroni post hoc test were used for comparison between multiple samples. Data are presented as mean +/− standard error of the mean (SEM) of at least three (for NSCLC) or representative of two (for neurons) independent experiments.

## 3. Results

### 3.1. Dynasore Blocks Transferrin Receptor Uptake and Ferroptosis

Upregulation of transferrin receptor TFR1 has been shown to promote ferroptosis [11,12]. TFR1 has long been used as bona-fide surface receptor in mechanistic studies on endocytosis. Dynamin 1 and 2 are known to mediate terminal membrane fission during clathrin-mediated endocytosis (CME) of transferrin receptor [21]. Therefore, we set out to test whether short-term dynamin-regulated TFR1 endocytosis would affect ferroptosis signaling. To set up a cellular system for this, we first tested ferroptosis sensitivity of a panel of KRAS-mutant lung cancer cell lines in response to erastin, which has been shown to inhibit xCT, thereby inducing ferroptosis [22].

Most cell lines tested showed a concentration-dependent decline in cell viability after exposure to erastin (Appendix A). Of note, the KRAS wild-type cell line H520 was as sensitive as many of the KRAS-mutant cell lines. Loss of viability could partially be rescued by the lipophilic antioxidant ferrostatin-1 (Fer-1), suggesting that these cells may be sensitive to ferroptotic cell death (Appendix A). Therefore, for further analysis, sensitive A549 and H441 cells were used. Importantly, erastin-induced cell death, as quantified by propidium iodide (PI) uptake, was entirely rescued by Fer-1 but not the pan-caspase inhibitor zVAD or the RIPK1 inhibitor necrostatin-1 (Nec-1s), excluding apoptotic and necroptotic cell death and pinpointing ferroptotic cell death to be induced by erastin in these cells (Figure 1A). The fact that Fer-1 treatment was not sufficient to entirely rescue loss of cell viability despite completely rescuing cell death prompted us to test whether apoptosis or necroptosis may contribute to loss of cell viability. However, neither Nec-1s nor zVAD or Fer-1 co-treatment entirely rescued loss of cell viability (Appendix A), suggesting that the residual cell viability loss was not cell death, but likely represented decreased cell proliferation in the presence of erastin. The erastin-sensitive cell line H441 expressed TFR1 (CD71) on the surface and siRNA-mediated suppression of dynamin 1 and 2 elevated surface expression of CD71 (Figure 1B), confirming a role for dynamin 1 and 2 in steady-state turnover of CD71 surface levels in these cells. Moreover, the dynamin 1 and 2 inhibitor dynasore [14] also increased surface levels of CD71 without significantly affecting cytosolic expression levels of CD71, supporting a functionality of dynamins in the endocytosis of CD71 in these cells (Figure 1C; Appendix A). Strikingly, co-treatment with dynasore entirely blocked ferroptosis induction by erastin or the GPX4 inhibitor RSL3 in a range of different cell lines (Figure 1D). Importantly, despite the fact that dynasore is commonly used at 80 µM to efficiently block dynamin 1 and 2, dynasore also potently inhibited ferroptosis at a range of lower concentrations (Appendix A). Lastly, in time-lapse imaging experiments, dynasore efficiently blocked RSL3 and erastin-induced ferroptosis, comparable to ferroptosis blockade by Fer-1, whereas zVAD did not affect cell death induction (Figure 1E). Taken together, these data identify dynasore as a highly effective inhibitor of ferroptosis in various cellular systems.

### 3.2. Inhibition of Dynamin 1- and 2-Regulated Iron Uptake is Insufficient to Block Ferroptosis

To validate whether dynasore-mediated inhibition of ferroptosis was mediated through its on-target activity against dynamin 1 and 2, we next performed siRNA-mediated silencing of dynamin 1 and 2 (Figure 2A). In order to validate that iron import was compromised by suppression of dynamin 1 and 2, we made use of the heavy metal indicator dye Phen Green SK diacetate (PG SK), of which the fluorescence has been shown to be quenched by intracellular labile iron pools [11,23]. As expected due to the fact that CD71 turnover was regulated by dynamin 1 and 2 in these cells (Figure 1B), suppression of dynamin 1 and 2 resulted in a loss of fluorescence quenching and thereby increased fluorescent signal, suggesting a decrease in intracellular labile iron pools (Figure 2B, Appendix A). Similarly, dynasore treatment also induced a comparable loss of fluorescent quenching, yet neither dynamin silencing nor dynasore treatment were as efficient as the iron-selective chelating agent DFO in decreasing intracellular iron pools (Figure 2B, right panel). However, despite decreasing intracellular iron pools, surprisingly, neither RSL3- nor erastin-induced cell death were rescued by dynamin 1 and 2 silencing (Figure 2C). Moreover, RSL3-induced lipid ROS accumulation was also not rescued by dynamin 1 and 2 silencing, demonstrating that in these cells dynamin-mediated short-term extracellular iron uptake is dispensable for ferroptosis execution (Figure 2D). These data strongly suggested that the on-target activity of dynasore against dynamin 1 and 2 and the resulting increased surface CD71 levels and decrease in intracellular iron were not sufficient to explain its strong ferroptosis inhibitory effect. Hence, these data pointed towards an additional off-target activity of dynasore that was responsible for potent ferroptosis inhibition. To next determine at which levels of the ferroptosis pathway dynasore may interfere, we evaluated a potential influence of dynasore on erastin-mediated reduction of cellular GSH. To this end we applied the fluorescent dye monochlorobimane (MCB), which reacts with thiols and therefore is widely used to selectively label GSH [24]. However, dynasore did not affect the reduction of GSH induced by erastin (Figure 2E), pointing towards dynasore regulating ferroptosis at a different level of the ferroptosis pathway. During ferroptosis, lipid ROS accumulation has been proposed to result in plasma membrane rupture [1]. Strikingly, RSL3- and erastin-induced accumulation of lipid ROS was entirely rescued by dynasore co-treatment (Figure 2F,G). These data indicated an additional off-target activity of dynasore between GSH depletion and enhanced lipid ROS formation that is ferroptosis protective. Therefore, dynasore-mediated on-target inhibition of dynamin 1 and 2 and modulation of the intracellular iron pool is insufficient to achieve blockade of ferroptosis and lipid ROS generation, and an additional off-target activity of dynasore had to be involved.

### 3.3. Dynasore Protects Neuronal Cells from Ferroptosis and Blocks Mitochondrial ROS Formation and Respiration

Dynasore has previously been shown to protect neurons from cell death after spinal cord injury in vivo [15]. However, the question of how dynasore may protect neurons from cell death has remained unexplored. Given that we found dynasore to block ferroptosis and ferroptosis is a cell death etiology linked to neuronal death, for example, in model systems of neurodegeneration in vitro [19,25] and in spinal cord injury in vivo [26], we next tested whether dynasore would also protect cells of neuronal origin from ferroptosis. Indeed, neuronal cells were sensitive to induction of ferroptosis by erastin and cell death could be reverted by rising concentrations of dynasore, albeit at the cost of a certain level of cell death induced by dynasore alone in these cells (Figure 3A). Supporting a functionality of dynasore in preventing ferroptosis also in neuronal cells, erastin-induced accumulation of lipid ROS was entirely reversed by co-treatment with dynasore (Figure 3B). Interestingly, apart from its activity on lipid ROS accumulation, we noted that dynasore also significantly blocked erastin-induced mitochondrial superoxide, as determined by mitoSOX staining (Figure 3C; Appendix A). Of note, mitochondrial superoxide is a known byproduct of oxidative phosphorylation. Therefore, we hypothesized that dynasore may affect mitochondrial superoxide production via inhibiting mitochondrial metabolic activity. To test this, we measured the metabolic activity of cells treated with rising concentrations of dynasore using their capacity to convert 3-(4,5-dimethylthiazol-2-yl)-2,5-diphenyltetrazolium bromide (MTT) to formazan. Indeed, dynasore inhibited cellular metabolic activity in a dose-dependent manner (Figure 3D). In addition, dynasore also reduced PG SK fluorescent quenching and thereby intracellular free iron levels over time in neuronal cells (Figure 3E,F), suggesting that this reduced iron availability may impact the respiratory chain and thereby mitochondrial ROS production. Indeed, dynasore treatment very effectively blocked basal cellular oxygen consumption rates (OCR) and maximal respiratory capacity, measured after adding the uncoupler carbonyl cyanide-p-trifluoromethyoxyphenylhydrazone (FCCP) to intact cells (Figure 3G). Similarly, extracellular acidification rates (ECAR) were strongly suppressed by dynasore (Figure 3H). Of note, another small molecule inhibitor against the dynamin family member dynamin-related protein 1 (Drp1), mdivi-1, was shown to block mitochondrial complex I-dependent oxygen consumption, and reverse electron transfer-mediated ROS production [27]. Therefore, in order to test whether dynasore may similarly directly affect complex I activity or mitochondrial respiration in general, we tested the effect of dynasore on the respiratory capacity of isolated mitochondria in cell-free systems, as was done previously [20]. However, in these cell-free systems, dynasore did not directly affect mitochondrial respiration (Figure 3I). In particular, the pronounced effects of dynasore on mitochondrial respiration in whole cells (Figure 3G) were neither achieved at the level of basal respiration nor at the other respiration states (Appendix A). Hence, in intact cells, other indirect mechanisms triggered by dynasore accounted for the observed metabolic effects.

These data suggested that the inhibition of mitochondrial ROS and respiration in intact cells may stem from dynasore-mediated reduced iron availability, rather than direct inhibition of mitochondrial complexes. Hence, dynasore indirectly suppresses mitochondrial respiration and ROS formation—a known requirement for the execution of erastin-induced ferroptosis in cellular assays [19,25,28]. However, the fact that dynasore also readily blocked RSL3-induced ferroptosis, and that this route of ferroptosis induction can also be exerted independently of the mitochondrial tricarboxylic acid cycle (TCA) and oxidative phosphorylation [28], suggested that there may be an additional activity of dynasore also preventing RSL3-induced ferroptosis independently of mitochondrial respiration inhibition and in addition to decreased iron levels.

### 3.4. Dynasore Functions as a Broadly Effective Radical-Trapping Agent

When testing what types of ROS dynasore may inhibit, we found that dynasore not only strongly inhibited overall cellular ROS induced by RSL3 or erastin treatment, but also significantly lowered the amount of basal ROS present in cells without RSL3 treatment (Figure 4A; Appendix A). Although some of this basal reduction in cellular ROS may stem from the inhibition of cellular respiration, the chemical structure of dynasore suggested the possibility that it may function as a direct ROS scavenger via its phenol or amine moiety [29]. Therefore, we next tested whether dynasore may directly scavenge ROS in cell-free systems. Indeed, dynasore reduced the stable radical 2,2-diphenyl-1-picrylhydrazyl (DPPH) almost as efficiently as trolox, a well-established antioxidant vitamin E derivative and blocker of ferroptosis (Figure 4B). Given its strong general ROS scavenging activity in cells and cell-free systems, we next tested whether dynasore may protect cells from cell death induced by hydrogen peroxide (H_2_O_2_), which unlike ferroptosis does not rely on lipid ROS formation or iron [1]. Strikingly, dynasore also potently blocked H_2_O_2_-induced cell death (Figure 4C,D). Importantly, while the general antioxidant N-acetylcysteine (NAC)—known to block H_2_O_2_-induced cell death—was very ineffective in blocking RSL3-induced ferroptosis (Figure 4E), dynasore was highly effective in blocking both types of cell death. Together, these data identify dynasore as a highly active pan-ROS cell death blocker which is equally potent in inhibiting cell death executed by lipid ROS as well as general ROS. Although dynasore decreases iron uptake through dynamin inhibition, this alone is insufficient to block ferroptosis. In addition, dynasore inhibits cellular respiration and general ROS production, which can feed into lipid peroxidation in the presence of iron. Lastly, dynasore acts as a direct radical scavenger that also blocks H_2_O_2_-induced cell death (summarized in Figure 4F).

In conclusion, we find that dynasore blocks pan-ROS-induced cell death by combined lowering of intracellular iron levels, inhibition of mitochondrial respiration and direct radical scavenging. These findings have important implications for the retrospective and future interpretation of studies observing the cell death-protective effects of dynasore. Moreover, our findings equally draw attention to the widely neglected possibility of an intrinsic antioxidant capacity of small molecules used to study the ferroptosis pathway, which should be taken into consideration for future studies making use of small molecules.

## 4. Discussion

Recently, ferroptosis was described as an iron-dependent form of regulated necrosis [1] (reviewed in [30,31]). Ferroptosis has been linked to mediating tissue injury in various human pathologies. For example, it has been suggested to be responsible for neurodegeneration of motor neurons, which occurs in a range of neurodegenerative conditions such as amyotrophic lateral sclerosis (ALS) [32,33] and Huntington´s disease [34]. Moreover, ferroptosis was demonstrated to mediate post-ischemic renal necrosis [5,18] and to participate in several other malignancies, including carcinogenesis, stroke, intracerebral hemorrhage and traumatic brain injury [34,35]. Here, we have identified a novel functionality of the small molecule inhibitor dynasore in blocking ferroptosis in neuronal and non-neuronal cells.

Dynasore was originally designed to target the small GTPases dynamin 1 and 2 [14], which regulate transferrin uptake by controlling earlier rate-limiting steps of clathrin-coated vesicle formation during endocytosis [10]. In line with this activity, dynasore treatment or suppression of dynamin 1 and 2 expression indeed led to surface accumulation of transferrin receptor (CD71). Although dynamins and dynasore indeed regulated intracellular iron levels, we surprisingly found that suppression of dynamin 1 and 2 expression did not block ferroptosis nor detrimental lipid ROS accumulation, whereas treatment with dynasore did. These data suggested that, at least in the short term, endocytosis of CD71 via dynamin 1 and 2 is less important for the execution of ferroptosis than previously thought. Of note, the iron chelating agent DFO, of which the capacity to block ferroptosis was decisive for naming this type of cell death, has been shown to act as a high affinity direct radical scavenger for peroxyl radicals in cell-free systems [36]. Based on the current model, in which peroxyl radicals propagate detrimental lipid peroxidation within membranes in ferroptotic cells [3], it is tempting to speculate that the lipid radical scavenging activity of DFO may be primarily responsible for blocking ferroptosis. Although transferrin in cell culture media FCS is vital for cells to undergo ferroptosis [12], our data suggest that long-term intracellular iron storage compartments and their mobilization may be more important for iron-fueled hydroxyl radical formation than endocytosis. Here, labile iron that was already available intracellularly or the activation of iron-dependent LOX may have been sufficient to trigger ferroptosis. Alternatively, the divalent metal-ion transporter-1 (DMT1), which has been widely described as a transferrin-independent iron transporter for non-hematopoietic cells, may have been sufficient to support ferroptosis in the cellular systems studied here [37,38].

One of the hallmarks of ferroptosis is the accumulation of lipid ROS that precedes instability and rupture of the plasma membrane [1]. Dynasore has been reported to exert biological effects associated with “off-target activity” which have not been explored so far [39]. Our data show that dynasore can counteract the accumulation of lipid ROS, as well as preventing the accumulation of mitochondrial and general cellular ROS by both suppressing mitochondrial respiration and acting as a radical trapping agent. ROS include hydroxyl or free radicals (^•^RO and ^•^OH), superoxide anions (O_2_^−^) and non-radical species such as hydrogen peroxide (H_2_O_2_) [1,40,41]. ROS are mainly produced as a result of mitochondrial metabolism and respiration, but they can also be released from cytosolic enzyme systems such as NADPH oxidases (NOX) [41]. Interestingly, hydroxyl radicals are generated from iron and H_2_O_2_ during oxygen metabolism via the Fenton and Haber–Weiss reactions [41]. Although we observed that dynasore blocks cellular respiration, which is a main source of cellular ROS, this inhibition was indirectly achieved, possibly through dynasore-mediated decreased iron uptake and a resulting decrease in mitochondrial iron availability to maintain respiratory chain activity. Interestingly, dynasore has been proposed as a candidate small molecule inhibitor to treat mitochondrial disorders characterized by aberrant accumulation of mitochondrial ROS [34,42]. Furthermore, in human corneal epithelial cells, dynasore prevented oxidative stress-induced cell damage through a non-described independent mechanism of endocytosis inhibition [43], and it is tempting to speculate as to whether the radical-trapping activity and mitochondrial respiratory inhibition discovered in our study may underlie such tissue protection. Although aberrant accumulation of ROS can cause DNA damage, protein denaturation and lipid peroxidation, certain amounts of iron and ROS are crucial for normal cell function [1,40,41]. This may explain the slight cytotoxic effects that dynasore treatment alone induces in our cellular systems. Although some studies have shown that erastin disrupts mPTP and induces apoptotic death of different cancer cells [44,45], our data clearly demonstrate that erastin induces ferroptosis in our cellular system and no other forms of regulated-cell death.

In vitro, cells exposed to exogenous H_2_O_2_ can trigger iron-dependent cell death through a process requiring lysosomal iron [46]. In contrast, Dixon et al. (2012) demonstrated that H_2_O_2_-induced death was not blockable by DFO, unlike RSL3- or erastin-induced ferroptosis [1]. These data suggest that while H_2_O_2_ can feed into lipid peroxidation via peroxyl radicals, this crosstalk to the ferroptosis pathway likely requires free labile iron to promote a Fenton reaction. Our data, showing that dynasore, which simultaneously lowers intracellular iron pools and functions as a direct radical scavenger, blocks ferroptosis and H_2_O_2_-induced cell death, supports this notion. Based on the observation that DFO does not rescue H_2_O_2_-induced cell death [1] and was shown to directly scavenge peroxyl radicals [36], we propose that dynasore does not specifically scavenge peroxyl radicals, but rather general radicals, explaining its inhibition of H_2_O_2_-induced cell death. In addition, we find that dynasore inhibits mitochondrial respiration, limiting cellular H_2_O_2_ while at the same time limiting iron availability. Although we find that decreased iron levels alone are not sufficient to block ferroptosis, we propose that these two activities together prevent Fenton chemistry, generating hydroxyl radical formation and lipid peroxidation, explaining its strong inhibition of ferroptosis.

## 5. Conclusions

Taken together, our findings propose two alternative mechanisms by which dynasore functions as a potent blocker of lipid ROS (ferroptosis) and ROS-driven cell death, respectively. First, combined inhibition of CD71-iron import and inhibition of mitochondrial respiration prevent lipid peroxidation and ferroptosis, and second, direct radical-trapping activity of dynasore blocks H_2_O_2_-induced cell death. These unexpected and novel findings suggest that Dynasore may be used as a new potent combined ROS blocker and lipid ROS-induced cell death inhibitor for in vivo studies. At the same time, our results suggest that urgent re-interpretation is warranted of studies making use of dynasore to block dynamin 1 and 2-mediated effects in vitro and in vivo.

## Figures and Tables

**Figure 1 cells-09-02259-f001:**
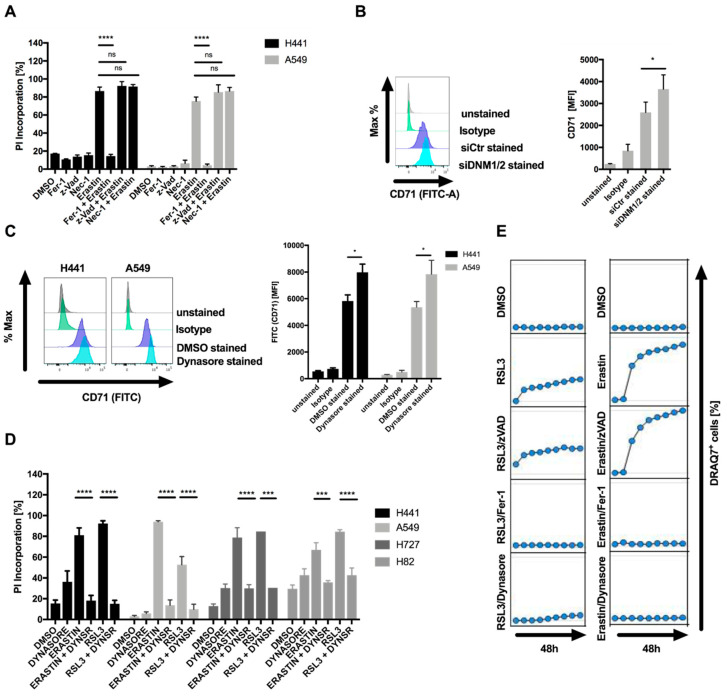
Dynasore inhibits ferroptosis. (**A**) The indicated cell lines were treated with DMSO, ferrostatin-1 (5 µM), erastin (10 µM), zVAD (20 µM), Nec-1 (20 µM) or the indicated combinations for 48 h. Cell death was quantified by propidium iodide (PI) uptake and flow cytometry. (**B**) H441 cells were transfected with control or dynamin 1 and 2 (DNM1-2)-targeting small interfering RNA (siRNA) for 48 h. Surface expression of CD71 was determined by staining and flow cytometry. (**C**) The indicated cells were treated with dynasore for 48 h. Surface expression of CD71 was quantified as in B. (**D**) Cells as indicated were treated and quantified as in A, but replacing ferrostatin-1 by dynasore (80 µM). (**E**) A549 cells were treated with DMSO, ferrostatin-1 (5 µM), erastin (10 µM), zVAD (20 µM), dynasore (80 µM) or the indicated combinations. Images were acquired every 8 h for 48 h. Dead cells were quantified as % DRAQ7 + cells using the IncuCyte image analysis software. All data are means +/− standard error of the mean (SEM) of at least three independent experiments, or representative images where applicable. MFI—mean fluorescence intensity. * indicates *p* < 0.05; ** indicates *p* < 0.01; *** indicates *p* < 0.001; **** indicates *p* < 0.0001; ns indicates non-significant differences.

**Figure 2 cells-09-02259-f002:**
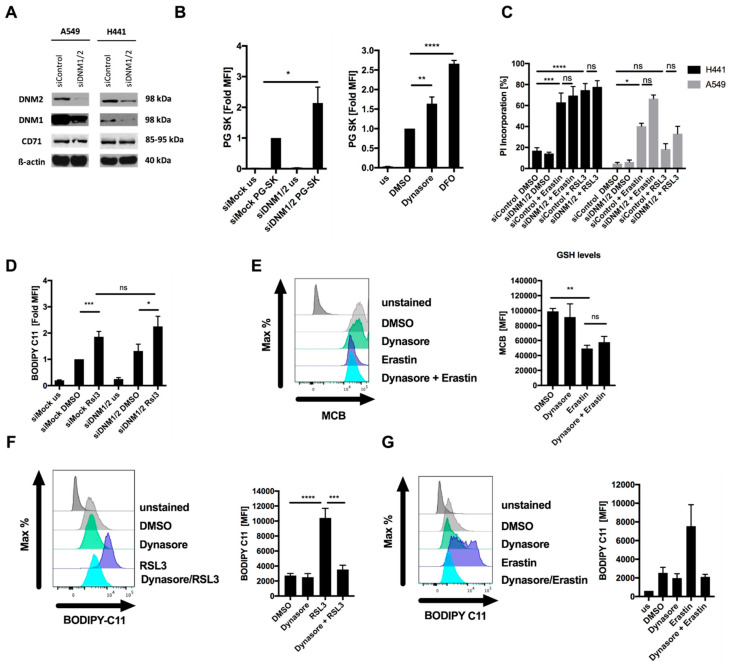
Dynasore blocks ferroptosis independently of dynamin 1 and 2. (**A**) Cells were subjected to control or dynamin 1 and 2 (DNM1-2)-targeting siRNA for 48 h, a representative blot is shown. (**B**) H441 cells (left panel) were stained by Phen Green SK staining (PG SK) or treated with dynasore (80µM) or DFO (100 µM) for 48 h (right panel). Fold mean fluorescence intensity (MFI) was determined by flow cytometry. (**C**) Cells as in A were treated with DMSO, erastin (10 µM) or RSL3 (1 µM) for 48 h. Cell death was determined by propidium iodide (PI) uptake and flow cytometry. (**D**) H441 cells were transfected with dynamin 1 and 2 (DNM1-2)-targeting siRNA for 48 h, followed by 5 h treatment with RSL3 (1µM). During the last 30 min, BODIPY C11 was added at 5 µM to each well. Mean fluorescence intensity (MFI) was quantified by flow cytometry. (**E**) H441 cells were treated with DMSO, erastin (10 µM), dynasore (80 µM) or both for 16 h and 48 h, respectively. During the last 30 min monochlorobimane (MCB) was added at 50 µM to each well. Mean fluorescence intensity (MFI) was quantified by flow cytometry. (**F**,**G**) H441 cells were treated with RSL3 (1µM) for 5 h or erastin (10 µM) for 16h. During the last 30 min BODIPY C11 was added at 5 µM to each well. Mean fluorescence intensity (MFI) was quantified by flow cytometry. All data are means +/− SEM of at least three independent experiments, or representative images where applicable. * indicates *p* < 0.05; ** indicates *p* < 0.01; *** indicates *p* < 0.001; **** indicates *p* < 0.0001; ns indicates non-significant differences.

**Figure 3 cells-09-02259-f003:**
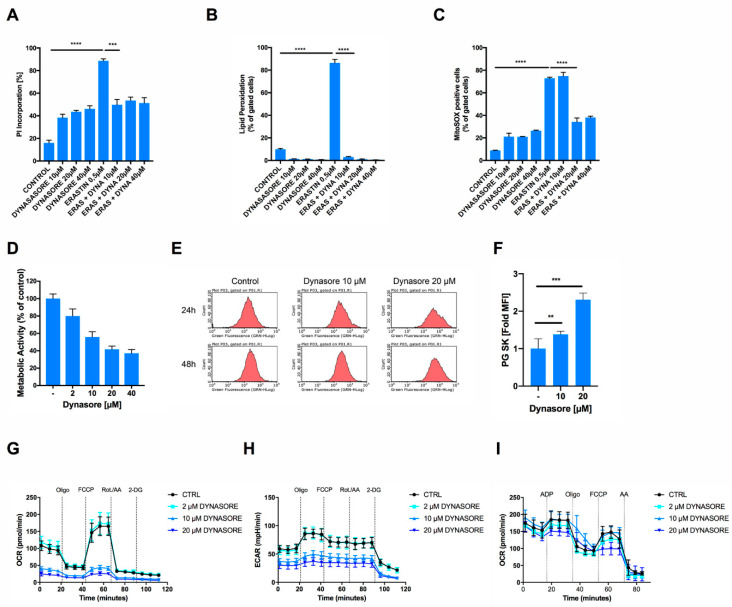
Dynasore protects neuronal cells from ferroptosis and inhibits mitochondrial respiration. (**A**,**B**) HT22 cells were treated with DMSO and with the indicated concentrations of dynasore and erastin (0.5 µM) for 16 h. Cell death was determined by AnnexinV/PI staining and fluorescence-activated cell sorting (FACS) measurements. Lipid peroxidation was quantified by BODIPY C11 staining and flow cytometry after 8 h of treatment. (**C**) HT22 cells were treated with DMSO, the indicated concentrations of dynasore and erastin (0.5 µM) for 16 h. MitoSOX-positive cells were gated and quantified by flow cytometry. (**D**) HT22 cells were incubated with the indicated concentrations of dynasore for 16 h. Metabolic activity was determined by MTT assay. (**E**,**F**) HT22 cells were treated as indicated for 24 h, stained by Phen Green SK staining (PG SK) (5 µM) and fold mean fluorescence intensity (MFI) was recorded by flow cytometry. (**G**,**H**) HT22 cells were treated with dynasore and erastin for 16 h. Measurement of the oxygen consumption rate (OCR) and extracellular acidification rate (ECAR) were determined simultaneously by seahorse assay. After recording of three baseline measurements, injections were performed as follows: (Oligo) 3 µM oligomycin, (FCCP) 0.5 µM FCCP, (Rot./AA) 100 nM Rotenone/1 µM Antimycin A, (2-DG) 50 mM 2-DG. (**I**) Sprague Dawley rat mitochondria were isolated, treated with dynasore for 1 h at the indicated concentrations and OCR was recorded by Seahorse Assay as in G. All data are means +/− SD or representative images where applicable. * indicates *p* < 0.05; ** indicates *p* < 0.01; *** indicates *p* < 0.001; **** indicates *p* < 0.0001; ns indicates non-significant differences.

**Figure 4 cells-09-02259-f004:**
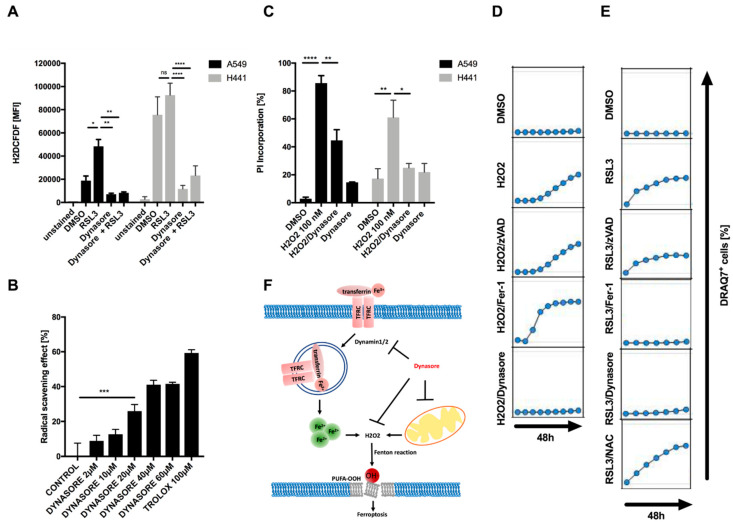
Dynasore functions as a broadly active radical scavenger. (**A**) Indicated cell lines were treated with DMSO, erastin (10 µM), dynasore (80 µM) or both for 16 h and 48 h, respectively. During the last 30 min H2DCFDF was added at 20 µM to each well. Mean fluorescence intensity (MFI) was quantified by flow cytometry. (**B**) Dynasore (2–60 µM) and trolox (100 µM) were incubated with DPPH (100 µM) for 30 min in the dark, then absorbance was measured at 517 nm to determine the radical scavenging activity. (**C**) Cells as in A were treated with DMSO, H_2_O_2_ (100 nM), Dynasore (80 µM) or both for 48 h. Cell death was determined by propidium iodide (PI) uptake and flow cytometry. (**D**,**E**) A549 cells were treated with DMSO, H_2_O_2_ (100 nM) or RSL3 (1 µM) alone or in combination with zVAD (20 µM), Fer-1 (5 µM), dynasore (80 µM) or NAC (1.5 mM). Images were acquired every 8 h for 48 h. Dead cells were quantified as % DRAQ7 + cells using the IncuCyte image analysis software. (**F**) Schematic overview of proposed mechanism of dynasore-mediated inhibition of ferroptosis. All data are means +/− SEM of three independent experiments or representative images where applicable. * indicates *p* < 0.05; ** indicates *p* < 0.01; *** indicates *p* < 0.001; **** indicates *p* < 0.0001; ns indicates non-significant differences.

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
