# Peer review of "Dynasore Blocks Ferroptosis through Combined Modulation of Iron Uptake and Inhibition of Mitochondrial Respiration"

_cells, 2020, doi:10.3390/cells9102259_

Round 1
Reviewer 1 Report
Clemente and colleagues investigated protective effects of dynasore, an inhibitor of the dynamin1/2, on the ferroptosis triggred by erastin and RSL3. This study is interesting; however, there are many serious concerns to accept these conclusions. As described in this manuscript, ferroptosis is well reported as a novel type of cell death, and erastin (or RSL3) is used to trigger this cell death. What stimuli induce this ferroptosis in vivo though authors describe that the ferroptosis is associated with various cell damages and tissues disorders (e.g., ALS)? Is this different from oxidative stress triggered by H2O2 and/or superoxide anion? GSH-depletion by treatment with BSO triggers the ferroptosis? Authors should describe these points in the Introductions or Discussion sections.
Major points;
1, Authors showed the amount of cell surface CD71 was elevated by dynasore and siRNA against dynamin1/2 (Fig. 1). These treatments increased intracellular CD71 proteins or not? In addition, intracellular iron was influenced in parallel?
2, Why did the treatment with dynasore alone induced cell death (Fig. 1H and Fig. 3)?
3, Authors showed that dynamin1/2-knockdown did not attenuate the erastin-induced cell death, ferroptosis (Fig. 2). Did the dynamin1/2-knockdown affect intracellular iron and erastin-induced ROS production? Authors describe that the transfecrrin pathway plays an important role in regulating intracellular iron.
4, In Fig. 3A, dynasore alone induced PI incorporation, cell death, though it completely abolished lipid peroxidation in Fig. 3B. Please explain this discrepancy.
5, What are differences between the erastin-and H2O2-induced cell death (Fig. 3F). Authors show that dynasore has an anti-oxidative property. Why didn’t the dynasore influence the H2O2-induced cell death at all?
6, In Fig. 4, authors performed the MTT assay in the presence or absence of dynasore. The treatment with dynasore decreased the MTT values in a dose-dependent manner. This means that dynasore is toxic against HT-22 cells though dynasore rescued erastin-induced cell death. Authors should describe this point in detail.
7, Authors showed that erastin attenuated mitochondrial activities. Dose erastin activate the canonical apoptotic pathway, caspase-9/-3? Author should clarify this point.
A Minor point;
1, There are lines around some figures. Authors had better delete them.
Author Response
Clemente and colleagues investigated protective effects of dynasore, an inhibitor of the dynamin1/2, on the ferroptosis triggred by erastin and RSL3. This study is interesting; however, there are many serious concerns to accept these conclusions. As described in this manuscript, ferroptosis is well reported as a novel type of cell death, and erastin (or RSL3) is used to trigger this cell death. What stimuli induce this ferroptosis in vivo though authors describe that the ferroptosis is associated with various cell damages and tissues disorders (e.g., ALS)? Is this different from oxidative stress triggered by H2O2 and/or superoxide anion? GSH-depletion by treatment with BSO triggers the ferroptosis? Authors should describe these points in the Introductions or Discussion sections.
We thank the reviewer for acknowledging the interest in our study. As requested, we have now included a discussion section on commonalities and differences in H2O2- and ferroptosis-mediated cell death in the revised version of the manuscript (discussion line 1015-1028). Overall, many very valuable points have been raised and addressing them has truly improved our study.
Major points;
1, Authors showed the amount of cell surface CD71 was elevated by dynasore and siRNA against dynamin1/2 (Fig. 1). These treatments increased intracellular CD71 proteins or not? In addition, intracellular iron was influenced in parallel?
We indeed find that dynamins regulate surface levels of CD71 but this does not significantly change total/intracellular protein level which is expected given that dynamin suppression prevents endocytosis of CD71 and thereby elevates surface levels (new supp. Figure S1D). We thank the reviewer for suggesting measurement of iron levels which has been very important for us to address. We have used Phen Green diacetate staining to determine free intracellular iron (as described in Yang…Stockwell 2008) and validated its quenching by free iron by the use of the iron-selective scavenger Deferoxamine (DFO) (new Figure 2B). Indeed, as suspected by the reviewer, we find that free iron levels are decreased upon dynamin1/2 suppression (as indicated by an increase in the fluorescent signal due to loss of fluorescent quenching). Yet, the effect is by no means as pronounced as with DFO. This was a very important point to raise and the data are added in new Figure 2B in the revised manuscript.
2, Why did the treatment with dynasore alone induced cell death (Fig. 1H and Fig. 3)?
Indeed, in some cell lines incubation with dynasore induces some background cell death as the reviewer rightly points out. Based on the new observation made following this reviewer’s suggestion that iron levels are decreased by dynasore and the fact that dynasore can strongly inhibit cellular respiration in intact cells (Figure 3G, H) , we propose that suppression of OXPHOS is responsible for the background levels of cell death observed. We have included a brief section about this point in the discussion (line 1008-1014).
3, Authors showed that dynamin1/2-knockdown did not attenuate the erastin-induced cell death, ferroptosis (Fig. 2). Did the dynamin1/2-knockdown affect intracellular iron and erastin-induced ROS production? Authors describe that the transfecrrin pathway plays an important role in regulating intracellular iron.
As the reviewer rightly points out, the fact that dynamin 1 and 2 silencing did not attenuate erastin-induced cell death was indeed unexpected. Therefore, to corroborate this finding we have now also tested dynamin 1 and 2 silencing and RSL3-induced cell death which can equally not be rescued by the knockdown (new Figure 2C). Importantly, we find that although dynamin 1 and 2 silencing does indeed lower intracellular iron levels (new Figure 2B), this is not sufficient to rescue cells from ferroptosis. In line with, this dynamin 1 and 2 silencing was also insufficient to rescue RSL3-induced lipid ROS generation (new Figure 2D) indicating that lowering iron import via the transferrin/dynamin route is insufficient to rescue from ferroptosis (a point we have extensively included in the discussion -see page line 896-903). These new data indicate that although dynamin 1 and 2 as well as dynasore can regulate surface CD71 and thereby affect intracellular iron levels, this is not sufficient to result in the strong ferroptosis inhibition achieved with dynasore pointing towards an additional activity of dynasore to be responsible for cell death inhibition (which we are uncovering in Figure 3 and 4).
4, In Fig. 3A, dynasore alone induced PI incorporation, cell death, though it completely abolished lipid peroxidation in Fig. 3B. Please explain this discrepancy.
This is an important point to clarify, as discussed above we propose that the dynasore-mediated background cell death is a result of inhibition of OXPHOS and does not involve ferroptosis (or lipid peroxidation). We now also observed that dynasore also decreases intracellular iron levels which are required for lipid peroxidation and hence may lower basal lipid ROS in these cells and explain the observation the reviewer raised. Therefore, we think that this cell death observed is due to inhibition of respiration and as such not a cell death driven by lipid peroxidation. Therefore, we see cell death in the absence of lipid peroxidation with dynasore alone. We now included a sentence on this in the discussion section to clarify this point (line 1008-1011).
5, What are differences between the erastin-and H2O2-induced cell death (Fig. 3F). Authors show that dynasore has an anti-oxidative property. Why didn’t the dynasore influence the H2O2-induced cell death at all?
Thank you for pointing this discrepancy, we have noted that the previous three killing experiments were performed at H2O2 concentration 10x too high leading to excessive ROS burden overriding any anti-oxidant activity. We have therefore repeated all H2O2 experiments (PI killing) in two cell lines each three times independently but also added time-lapse cell death analysis using the Incucyte imaging platform. And indeed, as the reviewer suspected, with the correct concentration of H2O2 we do see that dynasore blocks also H2O2-induced cell death (both in PI uptake assays and incucyte imaging analysis) which is, importantly, not blockable by the lipophilic antioxidant ferrostatin-1 (new Figure 4C, D E). Importantly, RSL3-induced ferroptosis was not efficiently blocked by the general antioxidant N-acetylcystein (NAC)-which is known to block H2O2-induced death- but very effectively by ferrostatin-1 and dynasore (new Figure 4E). Based on these data we propose that dynasore is a superior inhibitor for cell death types driven by ROS and lipid ROS as it a) lowers iron uptake which is required to fuel lipid ROS generation from H2O2 and b) directly scavenges ROS thereby blocking cell death driven by lipid ROS (ferroptosis) and general ROS not dependent on iron-fueled lipid peroxidation (H2O2). Again, we want to thanks this reviewer for pointing this out as addressing it truly helped the mechanistic understanding of how dynasore functions and we have now for clarification also included a mechanistic scheme of the mechanism by which dynasore blocks ferroptosis (new Figure 4F).
6, In Fig. 4, authors performed the MTT assay in the presence or absence of dynasore. The treatment with dynasore decreased the MTT values in a dose-dependent manner. This means that dynasore is toxic against HT-22 cells though dynasore rescued erastin-induced cell death. Authors should describe this point in detail.
The MTT assay measures metabolic activity which is distinct from a cell death/toxicity read-out. This is the reason why we are mostly using PI incorporation to determine membrane integrity as a cell death read-out. While dynasore can inhibit metabolic activity due to its inhibition of cellular respiration this does not immediately kill the cell as also seen by the levels of PI incorporation at the same dose. Therefore, throughout the manuscript we have included uptake of PI as a specific assay measuring membrane integrity and thereby cell death.
7, Authors showed that erastin attenuated mitochondrial activities. Dose erastin activate the canonical apoptotic pathway, caspase-9/-3? Author should clarify this point.
We agree that this was an important point to address. to clarify this, we have now added experiments co-treating erastin-treated cells with the pan caspase inhibitor zVAD or the RIPK1 inhibitor Nec-1 in comparison to co-treatment with Fer-1 in revised Figure 1A and caspase inhibition also in time-lapse studies in new Figure 1E. Neither the pan-caspase zVAD nor Nec-1s blocked erastin-induced cell death, indicating that erastin induces caspase-independent and non-necroptotic but in fact ferroptotic cell death in the cells studied
A Minor point;
1, There are lines around some figures. Authors had better delete them.
Thank you for pointing this out, this has been corrected in the revised version of the manuscript.
Reviewer 2 Report
The study by Clemente et al., report the effect of the endocytosis inhibitor Dynasore on ferroptosis in various cancer cells and and the HT22 neuronal cell line. Although the hypothesis is clear, the experimental setup and the conclusion of the authors are not supported by the presented data and the study has serious limitation. Below are some of my concerns:
- Ferroptosis is an iron-dependent cell death pathway that is different from apoptosis and necrosis. It can be triggered by depletion of GSH through erastin or cysteine-free media or inhibition of GPX4 by RSL3 or other methods. However, GSH is not only important to ferroptosis and its depletion may lead to oxidative stress and apoptosis. Erastin has a well-known effect on VDAC in mitochondria and its use at high concentrations and over a prolonged period may lead to non-ferroptotic death. The authors have correctly used a short time 16h to measure lipid peroxidation (BODIPY) and other ROS markers but used a very long period 48h for toxicity assays. They should have explored a time dependent cell death with and without ferrostatin and show when cell death is ferroptotic and when it is not. Erastin toxicity in Fig. 1a does not show a typical dose-dependent cell death and most of the cell viability is lost at very high concentration 3-10 uM and some cells didn’t die at those concentration. The fact that ferrostatin only partially rescued this loss in viability demonstrate that there is an effect on cell proliferation and possibly apoptosis under those conditions. Those effects should have been investigated thoroughly by the use of apoptosis inhibitors and time curse and not using PI staining at very long incubation times.
- Other studies have shown that erastin induced apoptosis and ferroptosis in cancer cell lines (Huang et al., Oncology reports 2018, Huo et al., PlosOne 2016)
- The authors claim that the mechanism of dynasore is not well understood but I’m aware of studies that have shown that dynasore suppresses cell proliferation, which is very relevant to this study including a mechanism of induction of cell arrest in G0/G1 cycle which may explain some of the effects observed here (Zhong et al., cell death&Disease 2019).
- The concentrations and time of incubations used here are very high and dynasore most likely affects several pathways with an effect on iron homeostasis, cell proliferation, cell division, apoptosis and the authors should investigate those pathways.
- The effect of dynasore and dynamin silencing on TfR1 uptake is not sufficiently elaborated. The authors should have used intracellular iron/ferritin to demonstrate a reduction of iron uptake as a result and provide a quantification estimate. The FACS data do not allow a proper evaluation of the effect on all cells. Surface biotinylation and ratio of cytosol to membrane would be also a good method. In supplementary fig.1 (uncropped blot) dynasore lead to an increase in intracellular TFR1 over 3, 6, 12 and 24h while at 48h it returned to normal levels, the authors did not investigate this effect and concluded no change.
- The effect of dynamin 1 and dynamin 2 silencing on erastin sensitivity is unexpected. If the silencing was effective as demonstrated by the westernblot then it should have been effective in reducing erastin toxicity but the opposite is observed dynamin 1/2 silencing increased toxicity (even if ns). Transferrin is required for erastin toxicity (Gao et al., Mol Cell 2015) and this points to a non-ferroptotic cell death by erastin
- The radical scavenging assay is not conclusive as it is more suited for water soluble radical scvengers (TROLOX is a water soluble verivative of vitamin E) and not lipid antioxidants
- The effect on mitochondria can not explain the RSL3 rescue by dynasore as mitophagy protects from erastin mediated ferroptopsis but not RSL3 mediated ferroptosis
Author Response
The study by Clemente et al., report the effect of the endocytosis inhibitor Dynasore on ferroptosis in various cancer cells and and the HT22 neuronal cell line. Although the hypothesis is clear, the experimental setup and the conclusion of the authors are not supported by the presented data and the study has serious limitation. Below are some of my concerns:
- Ferroptosis is an iron-dependent cell death pathway that is different from apoptosis and necrosis. It can be triggered by depletion of GSH through erastin or cysteine-free media or inhibition of GPX4 by RSL3 or other methods. However, GSH is not only important to ferroptosis and its depletion may lead to oxidative stress and apoptosis. Erastin has a well-known effect on VDAC in mitochondria and its use at high concentrations and over a prolonged period may lead to non-ferroptotic death. The authors have correctly used a short time 16h to measure lipid peroxidation (BODIPY) and other ROS markers but used a very long period 48h for toxicity assays. They should have explored a time dependent cell death with and without ferrostatin and show when cell death is ferroptotic and when it is not. Erastin toxicity in Fig. 1a does not show a typical dose-dependent cell death and most of the cell viability is lost at very high concentration 3-10 uM and some cells didn’t die at those concentration. The fact that ferrostatin only partially rescued this loss in viability demonstrate that there is an effect on cell proliferation and possibly apoptosis under those conditions. Those effects should have been investigated thoroughly by the use of apoptosis inhibitors and time curse and not using PI staining at very long incubation times.
We thank the reviewer for pointing this out. To address these points, we have included co-incubation experiments with the pan-caspase inhibitor zVAD to test for an apoptotic contribution as well as with the RIPK1 inhibitor Necrostatin-1 to test for a necroptotic contribution in erastin-induced cell death again using PI incorporation as the read-out to focus only on cell death induction and excluding proliferative effects. We find that indeed, in our models used, erastin only triggers cell death blockable by the lipophilic radical scavenger ferrostatin-1 but not by any of the other inhibitors (new Figure 1A). In order to not only test one time point but resolve cell death over time, an important point to raise, we have also included time-lapse analysis of cell death kinetics with RSL3 and erastin +/- Fer-1 or zVAD. In both cases, cell death induced is only blocked by Fer-1 (or dynasore) but not zVAD pinpointing this to be ferroptotic cell death in these cells (new Figure 1E). Moreover, to address the concern regarding viability not being fully rescued by Fer-1, we have also included viability experiments +/- Fer-1, zVAD, or Nec-1s in new Supp. Figure 1C. None of these cell death inhibitors fully rescued viability (despite Fer-1 fully rescuing cell death, Figure 1A) proposing that the remaining loss of viability does not represent cell death but decreased proliferation as expected from erastin which can also target VDACs. Thereby, we can exclude a role for caspase-dependent apoptosis and caspase-inhibited necroptosis in these cells.
While we agree that erastin clearly has additional effects on proliferation in these cells due to other targets as can be seen in viability assays, we have focused on erastin-induced ferroptotic cell death within the scope of this study which is entirely blockable by co-treatment with ferrostatin-1 in a selective cell death read-out such as uptake of propidium iodide (PI). Together with the time-lapse cell death kinetic experiments we believe that this should now sufficiently establish specific ferroptosis induction.
- Other studies have shown that erastin induced apoptosis and ferroptosis in cancer cell lines (Huang et al., Oncology reports 2018, Huo et al., PlosOne 2016)
We have now included these citations also into the discussion to also point these activities of erastin out (line 1011-1012). However, as we have now extensively tested a potential role for caspase activity (see reply above) we can conclude that caspase activity does not participate in erastin-induced cell death in our cellular systems (see also new Figure 1A, E; Supp. Figure 1C, new Figure 4E). These data suggest that the extent to which erastin triggers caspase activation may vary between different types of cellular systems used.
- The authors claim that the mechanism of dynasore is not well understood but I’m aware of studies that have shown that dynasore suppresses cell proliferation, which is very relevant to this study including a mechanism of induction of cell arrest in G0/G1 cycle which may explain some of the effects observed here (Zhong et al., cell death&Disease 2019).
We apologize as we may not have been clear in the text and have rephrased, the mechanisms by which dynasore protects from tissue injury and cell death are only poorly understood. We have changed this in the abstract (line 29).
- The concentrations and time of incubations used here are very high and dynasore most likely affects several pathways with an effect on iron homeostasis, cell proliferation, cell division, apoptosis and the authors should investigate those pathways.
We agree that 80µM may seem high. However, this is the standard dose used by most studies which make use of dynasore as an inhibitor to block dynamin 1 and 2 in intact cells including the initial study identifying it as a dynamin 1 and 2 inhibitor (Macia et al. 2006). However, to address exactly this point, we have also titrated dynasore down to 20µM and still observe equally significant blockade of ferroptosis (Supp. Figure 1E). We thank the reviewer for suggesting measurement of iron levels which has been very important for us to address. We have used Phen Green diacetate staining to determine free intracellular iron (as described in Yang…Stockwell 2008) and validated its quenching by free iron by the use of the iron-selective scavenger Deferoxamine (DFO) (new Figure 2B). Indeed, as suspected by the reviewer, we find that free iron levels are decreased upon dynamin1/2 suppression (as indicated by an increase in the fluorescent signal due to loss of fluorescent quenching). Yet, the effect is by no means as pronounced as with DFO. This was a very important point to raise and the data are added in new Figure 2B in the revised manuscript.
- The effect of dynasore and dynamin silencing on TfR1 uptake is not sufficiently elaborated. The authors should have used intracellular iron/ferritin to demonstrate a reduction of iron uptake as a result and provide a quantification estimate. The FACS data do not allow a proper evaluation of the effect on all cells. Surface biotinylation and ratio of cytosol to membrane would be also a good method. In supplementary fig.1 (uncropped blot) dynasore lead to an increase in intracellular TFR1 over 3, 6, 12 and 24h while at 48h it returned to normal levels, the authors did not investigate this effect and concluded no change.
We thank the reviewer for this constructive criticism and agree that investigations along these lines were vital for the conclusions of the study. As suggested we have now included assays to determine relative amounts of intracellular free iron pools (see answer to 4.) In brief, both, silencing of dynamin 1 and 2 and dynasore did resulted in decreased intracellular iron, yet this was insufficient to block cell death or lipid peroxidation induced by RSL3 (new Figure 2B-D).Since these effects on CD71 and iron import were insufficient to block ferroptosis (as concluded from the observation that silencing of dynamin1/2 increased surface CD71 in turn reduced intracellular iron but did not block ferroptosis- see Figure 1B, Figure 2B-D), we further focused on other additional effects which may be responsible for ferroptosis blockade by dynasore within the scope of this study.
- The effect of dynamin 1 and dynamin 2 silencing on erastin sensitivity is unexpected. If the silencing was effective as demonstrated by the westernblot then it should have been effective in reducing erastin toxicity but the opposite is observed dynamin 1/2 silencing increased toxicity (even if ns). Transferrin is required for erastin toxicity (Gao et al., Mol Cell 2015) and this points to a non-ferroptotic cell death by erastin
We agree that the fact that dynamin 1 and 2 do not regulate ferroptosis was unexpected and also surprising to us when we found it but to corroborate this finding we have now also included data on RSL3-mediated ferroptosis and dynamin 1 and 2 silencing showing a similar outcome. This indeed pertains to our key finding: While dynasore surely inhibits dynamin 1 and 2 and, with that, also increases CD71 surface levels and decrease intracellular iron, this on-target activity is not sufficient to achieve ferroptosis inhibition pointing at an additional off-target effect of dynasore being responsible for the strong ferroptosis inhibition observed. We are aware that transferrin was shown to regulate ferroptosis, a publication which we also cite and discuss and propose that short-term dynamin-mediated endocytosis of CD71 is not required for iron- bursts mobilized during acute ferroptosis which may stem from intracellular stores. This is not to say that long-term deletion of CD71 or dynamins will deplete also these intracellular stores. This point is now extensively discussed (line 901-906).
- The radical scavenging assay is not conclusive as it is more suited for water soluble radical scvengers (TROLOX is a water soluble verivative of vitamin E) and not lipid antioxidants
Thank you for pointing out this discrepancy regarding whether dynasore acts as a lipophilic- (ferroptosis) or general radical scavenger, we have noted that the previous H2O2 three killing experiments were performed at H2O2 concentration 10x too high leading to excessive ROS burden overriding any anti-oxidant activity. We have therefore repeated all H2O2 experiments (PI killing) in two cell lines each three times independently but also added time-lapse cell death analysis using the Incucyte imaging platform. With the correct concentration of H2O2 we do see that dynasore blocks also H2O2-induced cell death (both in PI uptake assays and incucyte imaging analysis) which is, importantly, not blockable by the lipophilic antioxidant ferrostatin-1 (new Figure 4C, D E). Importantly, RSL3-induced ferroptosis was not efficiently blocked by the general antioxidant N-acetylcystein (NAC)-which is known to block H2O2-induced death- but very effectively by ferrostatin-1 and dynasore (new Figure 4E). Based on these data we propose that dynasore is a superior inhibitor for cell death types driven by ROS and lipid ROS as it a) lowers iron uptake which is required to fuel lipid ROS generation from H2O2 and b) directly scavenges water-soluble radicals thereby blocking cell death driven by lipid ROS (ferroptosis) and general ROS not dependent on iron-fueled lipid peroxidation (H2O2). Again, we want to thanks this reviewer for pointing this out as addressing it truly helped the mechanistic understanding of how dynasore functions and we have now for clarification also included a mechanistic scheme of the mechanism by which dynasore blocks ferroptosis (new Figure 4F).
- The effect on mitochondria can not explain the RSL3 rescue by dynasore as mitophagy protects from erastin mediated ferroptopsis but not RSL3 mediated ferroptosis
While we agree with this statement for some cell types (Gao et al. 2019), in other cell types mitochondria can be involved also in RSL3-induced ferroptosis (Jelinek at el. 2018), a statement supported also by our data showing that general ROS is not increased in all cell lines we tested upon RSL3 treatment (new Figure 4A). Irrespective of cell type, we think that the general chemical radical scavenging property of dynasore in addition to lowering free iron available for Fenton chemistry and generation of hydroxyl radicals and lipid ROS is responsible for inhibiting RSL3-induced ferroptosis. In support of this argument, we also observe that dynasore lowers basal lipid ROS in neuronal cells new (Figure 3B).
Reviewer 3 Report
The present study by Clemente et al. reports the inhibitory effect of dynasore, a cell-permeable chemical inhibitor of Dynamin1/2, on ferroptosis. The manuscript is well-written and the conclusions are solid. The authors provide a strong case for the existence of off-target activities of dynasore, as a direct scavenger of ROS and as a compound that targets mitochondria. I have the following suggestions/questions to improve the manuscript.
- In Fig 3E, the authors examine the effect of dynasore on the chemical DPPH. This cell-free assay is presented as a proof that dynasore is directly anti-oxidant. Can the authors be more precise regarding the nature of the ROS that are scavenged by dynasore: superoxide, hydroxyle?
- Fig. 2B : in A459 cells, the authors show an almost two-fold difference in ferroptosis between the siControl and siDNM1/2 conditions. Despite the small error bars, the authors claim that the difference is non-significant. Can the authors confirm that it is indeed the case and explain how they performed their statistical analyses?
- The study aims to document the different activities through which dynasore blocks ferroptosis. In this respect, I think that the manuscript lacks an experimental section with a direct analysis of the effect of dynasore on the cellular iron pool and the metabolism of iron (ferritinophagy ?). Otherwise, the discussion on the respective role of CD71 endocytosis vs ROS chelation and mitochondrial modulation is merely speculative and poorly supported by the data (line 345).
Author Response
The present study by Clemente et al. reports the inhibitory effect of dynasore, a cell-permeable chemical inhibitor of Dynamin1/2, on ferroptosis. The manuscript is well-written and the conclusions are solid. The authors provide a strong case for the existence of off-target activities of dynasore, as a direct scavenger of ROS and as a compound that targets mitochondria. I have the following suggestions/questions to improve the manuscript.
We thank the reviewer for acknowledging the quality and interest in our study.
- In Fig 3E, the authors examine the effect of dynasore on the chemical DPPH. This cell-free assay is presented as a proof that dynasore is directly anti-oxidant. Can the authors be more precise regarding the nature of the ROS that are scavenged by dynasore: superoxide, hydroxyle?
We thank the reviewer for raising this point, although we cannot exclude hydroxyl radicals we propose that dynasore scavenges more general types of ROS as it also blocked H2O2-induced cell death (new Figure 4C, D). Yet, unlike NAC it potently blocks ferroptosis which may be related to its function in lowering intracellular free iron, which alone is insufficient to block cell death in these cells (new Figure 2B-D). Based on these new data, we propose a model in which dynasore scavenges general types of ROS (blocking H2O2-induced death), lowers intracellular iron (alone is insufficient to block ferroptosis) and inhibits mitochondrial ROS production which, in the presence of iron can feed into hydroxyl radical formation via the Fenton reaction. This model is now also included as a scheme in new Figure 4F.
- Fig. 2B : in A459 cells, the authors show an almost two-fold difference in ferroptosis between the siControl and siDNM1/2 conditions. Despite the small error bars, the authors claim that the difference is non-significant. Can the authors confirm that it is indeed the case and explain how they performed their statistical analyses?
We used two-way ANOVA with Bonferroni post test to determine differences and data are shown as means +/- SEM (giving small error bars, despite not being significant). While the same analysis may turn out to be significant using a t-test, this will not change the conclusion of the figure which demonstrates that dynamin 1 and 2 silencing does not rescue erastin or RSL3-induced cell death (if anything, silenced cells die more, now also including the same experiment with RSL3-treated cells in new Figure 2C).
- The study aims to document the different activities through which dynasore blocks ferroptosis. In this respect, I think that the manuscript lacks an experimental section with a direct analysis of the effect of dynasore on the cellular iron pool and the metabolism of iron (ferritinophagy ?). Otherwise, the discussion on the respective role of CD71 endocytosis vs ROS chelation and mitochondrial modulation is merely speculative and poorly supported by the data (line 345).
We thank the reviewer for suggesting measurement of iron levels which has been very important for us to address. We have used Phen Green diacetate staining to determine free intracellular iron (as described in Yang…Stockwell 2008) and validated its quenching by free iron by the use of the iron-selective scavenger Deferoxamine (DFO) (new Figure 2B). Indeed, as suspected by the reviewer, we find that free iron levels are decreased upon dynamin1/2 suppression (as indicated by an increase in the fluorescent signal due to loss of fluorescent quenching). Similarly, dynasore reduces intracellular iron pools as expected due to it on-target activity on dynamin-mediated CD71 endocytosis. Yet, the effect is by no means as pronounced as with DFO and insufficient to rescue from ferroptosis (Figure 2C) and lipid peroxidation (new Figure 2D). This was a very important point to raise and the data are added in the revised manuscript.
Round 2
Reviewer 2 Report
The manuscript has been substantially improved and the authors have presented a solid revision which has consolidates some of the previous finding while at the same added additional information such as the effect of Dynasore on iron uptake in the context of ferroptosis.
I am satisfied with the presented data and I have no further comments
Reviewer 3 Report
The authors have addressed my comments
I have no other point to raise.